# FOXM1 Is a Novel Molecular Target of AFP-Positive Hepatocellular Carcinoma Abrogated by Proteasome Inhibition

**DOI:** 10.3390/ijms23158305

**Published:** 2022-07-27

**Authors:** Ru Li, Hikari Okada, Taro Yamashita, Kouki Nio, Han Chen, Yingyi Li, Tetsuro Shimakami, Hajime Takatori, Kuniaki Arai, Yoshio Sakai, Tatsuya Yamashita, Eishiro Mizukoshi, Masao Honda, Shuichi Kaneko

**Affiliations:** Department of Gastroenterology, Kanazawa University Hospital, Kanazawa 920-8641, Japan; staceylee1221@gmail.com (R.L.); okada0922@gmail.com (H.O.); nio@m-kanazawa.jp (K.N.); chenhan19890801@wchscu.cn (H.C.); liyingyi@staff.kanazawa-u.ac.jp (Y.L.); shimakami@m-kanazawa.jp (T.S.); takatori@m-kanazawa.jp (H.T.); arai@m-kanazawa.jp (K.A.); yoshios@m-kanazawa.jp (Y.S.); ytatsuya@m-kanazawa.jp (T.Y.); eishirom@m-kanazawa.jp (E.M.); mhonda@m-kanazawa.jp (M.H.); skaneko@m-kanazawa.jp (S.K.)

**Keywords:** alpha-fetoprotein, forkhead box M1, carfilzomib, hepatocellular carcinoma, vascular endothelial growth factor receptor 2

## Abstract

Alpha-fetoprotein (AFP) is an oncofetal protein that is elevated in a subset of hepatocellular carcinoma (HCC) with poor prognosis, but the molecular target activated in AFP-positive HCC remains elusive. Here, we demonstrated that the transcription factor forkhead box M1 (FOXM1) is upregulated in AFP-positive HCC. We found that FOXM1 expression was highly elevated in approximately 40% of HCC cases, and FOXM1-high HCC was associated with high serum AFP levels, a high frequency of microscopic portal vein invasion, and poor prognosis. A transcriptome and pathway analysis revealed the activation of the mitotic cell cycle and the inactivation of mature hepatocyte metabolism function in FOXM1-high HCC. The knockdown of FOXM1 reduced AFP expression and induced G2/M cell cycle arrest. We further identified that the proteasome inhibitor carfilzomib attenuated FOXM1 protein expression and suppressed cell proliferation in AFP-positive HCC cells. Carfilzomib in combination with vascular endothelial growth factor receptor 2 (VEGFR2) blockade significantly prolonged survival by suppressing AFP-positive HCC growth in a subcutaneous tumor xenotransplantation model. These data indicated that FOXM1 plays a pivotal role in the proliferation of AFP-positive liver cancer cells. Carfilzomib can effectively inhibit FOXM1 expression to inhibit tumor growth and could be a novel therapeutic option in patients with AFP-positive HCC who receive anti-VEGFR2 antibodies.

## 1. Introduction

Hepatocellular carcinoma (HCC) is one of the most common malignancies with a dismal outcome [1,2]. Recent progress in molecular oncology has enabled HCC patients to receive systemic therapies targeting immune checkpoint molecules (cytotoxic T-lymphocyte-associated protein 4 with ipilimumab, programmed cell death 1 with nivolumab, and programmed cell death 1 ligand 1 with atezolizumab), angiogenesis (vascular endothelial growth factor receptor 2 (VEGFR2) with ramucirumab, sorafenib, regorafenib, lenvatinib, and cabozantinib), and receptor tyrosine kinase pathways (Raf with sorafenib and regorafenib, fibroblast growth factor receptor with lenvatinib, and c-Met with cabozantinib), which has dramatically extended the overall survival of advanced stage patients [3]. Nevertheless, HCC in these patients generally acquires resistance to treatment and shows disease progression [4]. The development of novel therapeutic strategies is required to provide alternative treatment options for these patients.

Alpha-fetoprotein (AFP) is a typical oncofetal protein that belongs to the albumin superfamily and is activated in hepatic progenitors and HCC cells [5]. AFP is expressed in 30–60% of HCC cases and its presence is generally correlated with a poor survival outcome. Previously, we demonstrated that AFP is not merely a tumor marker but also reflects the malignant nature of HCC with stem cell features [6,7]. Recently, a monoclonal antibody targeting VEGFR2 was shown to effectively prolong the overall survival of advanced stage HCC patients with elevated serum AFP levels [8,9]. However, the mechanistic link between AFP production and VEGFR2 expression is unclear. Furthermore, suppression of the angiogenesis pathway alone is not sufficient to inhibit the growth of AFP-positive HCC because it only prolongs the median overall survival by approximately 1.2 months [9]. Therefore, the identification of novel molecular targets is critical for developing unique treatment options in advanced AFP-positive HCC.

Forkhead box proteins are a family of evolutionarily conserved transcriptional regulators [10,11]. Among them, forkhead box M1 (FOXM1) is associated with the activation of the mitotic program and is expressed in proliferating cells [12,13]. FOXM1 stimulates the expression of genes involved in various steps of tumor progression, including epithelial–mesenchymal transition, cell migration, and premetastatic niche formation [14]. Previously, we demonstrated that FOXM1 is upregulated in AFP-positive HCC [15], suggesting that FOXM1 may be a molecular target if therapeutic abrogation is available.

In the present study, we evaluated the role of FOXM1 expression on signaling pathways, cancer stem cell marker status, and survival outcome in HCC patients. We further evaluated the effect of carfilzomib, a proteasome inhibitor suppressing FOXM1 protein that we identified in this study, in combination with a monoclonal antibody targeting VEGFR2 on HCC growth.

## 2. Results

### 2.1. FOXM1 Expression and Prognosis in Surgically Resected HCC

We evaluated FOXM1 expression in 133 HCC tissues surgically resected at Kanazawa University Hospital (Cohort 1). Patients in this cohort included HCV-related cases with relatively older ages, and approximately half of these patients did not develop liver cirrhosis (Table 1).

The FOXM1 expression status was varied in each HCC tissue, and FOXM1 was located in the nucleus (Figure 1A). Of these HCC cases, 54 (~40%) were classified as FOXM1-high (>25% of nuclei stained with anti-FOXM1 antibody) and 79 (~60%) were defined as FOXM1-low (≤25% of nuclei stained). We evaluated the clinicopathological characteristics of the FOXM1-high and -low HCC cases, and FOXM1-high HCC was significantly associated with high serum AFP levels, poorly differentiated histological findings, large tumor size, and a high frequency of microscopic portal vein invasion (Figure 1B and Table 2). We evaluated the recurrence-free (Figure 1C) and overall (Figure 1D) survival of these HCC patients and found that FOXM1-high HCC patients showed worse recurrence-free (*p* = 0.0013) and overall (*p* = 0.018) survival compared with FOXM1-low HCC patients.

### 2.2. Transcriptomic Characteristics of FOXM1-High HCC

To evaluate the molecular profiles of FOXM1-high HCC, we analyzed an Affymetrix gene expression dataset of 238 primary HCC tissues with available serum AFP information (Cohort 2). Patients in this cohort contained only HBV-related cases with relatively younger ages, and most of the patients had a cirrhotic liver (Table 1). Ninety-five (~40%) and 143 (~60%) HCC cases were regarded as FOXM1-high and -low HCC, respectively, based on the frequency of FOXM1-high HCC evaluated by immunohistochemistry. We evaluated the clinicopathological characteristics of FOXM1-high and -low HCC, and again, FOXM1-high HCC was associated with high serum AFP levels (Table 2). FOXM1-high HCC was also significantly associated with macroscopic tumor thrombosis and advanced BCLC stages.

We performed a class comparison analysis with *t*-tests and permutation tests (*p* < 0.001) of the class labels (FOXM1-high and -low) using BRB-ArrayTools (version 4.3.2) and identified 2119 genes differentially expressed between the classes (1275 genes upregulated and 844 downregulated in FOXM1-high HCC compared with FOXM1-low HCC) (Figure 2A). Among them, typical liver cancer stem cell markers such as AFP and keratin 19 (*KRT19*) were upregulated in FOXM1-high HCC, whereas typical mature hepatocyte markers such as solute carrier organic anion transporter family member 1B1 (*SLCO1B1*) and cytochrome P450 3A4 (*CYP3A4*) were downregulated (Figure 2B). A weak positive correlation was observed between FOXM1 and AFP signal intensities in 238 microarray samples (Figure 2C, r = 0.37, *p* < 0.0001, Spearman’s rank correlation coefficient). We performed pathway analysis using these gene sets by MetaCore software (http://portal.genego.com (accessed on 26 May 2020)). Noticeably, the pathways activated in FOXM1-high HCC were strongly associated with mitotic cell cycle processes (Cluster A), whereas those inactivated were strongly associated with mature hepatocyte metabolism (Cluster B) (Figure 2D).

We evaluated the recurrence-free (Figure 2E) and overall (Figure 2F) survival of Cohort 2 HCC patients and found that FOXM1-high HCC patients showed worse recurrence-free (*p* = 0.059) and overall (*p* = 0.043) survival compared with FOXM1-low HCC patients.

Taken together, the above data indicated that FOXM1 was upregulated in approximately 40% of HCC cases with serum AFP elevation and poor survival outcome, and its overexpression was strongly correlated with the activation of stem cell markers and the mitotic cell cycle and the inactivation of mature hepatocyte metabolism function.

### 2.3. FOXM1 Inhibition Represses the Cell Cycle

Because FOXM1 activation in HCC was associated with serum AFP elevation, mitotic cell cycle activation, and poor prognosis, we tested whether FOXM1 could be a molecular target in AFP-positive HCC. We knocked down FOXM1 gene expression using siRNAs in Huh7 cells (Figure 3A, *p* < 0.05). Western blot showed the reduction in FOXM1 protein using the same condition (Figure 3B). Interestingly, FOXM1 knockdown resulted in a reduction in AFP gene expression in Huh7 cells (Figure 3C, *p* < 0.05). FOXM1 knockdown also inhibited cell proliferation (Figure 3D, *p* < 0.05), consistent with the reported role of FOXM1 as a regulator of the cell cycle. We tested the effect of FOXM1 knockdown on the cell cycle and found that FOXM1 knockdown increased the number of G2-phase cells and decreased the number of G1/S-phase cells, indicating that FOXM1 knockdown induced G2/M cell cycle arrest (Figure 3E). These data suggested that the transcription factor FOXM1 might play a role even in the M phase, when most transcription factors are evicted from chromosomes. We evaluated the subcellular localization of FOXM1 in M-phase and interphase Huh7 cells by immunofluorescence. Because fluorescence microscope images could not be used to precisely evaluate the location of FOXM1, we utilized super-resolution microscopy to visualize its location on DNA at the nanometer scale. FOXM1 was ubiquitously detected on interphase chromosomes as dots (Figure 3F, upper panels). Interestingly, although FOXM1 was mainly dispersed in the cytoplasm at the M phase as dots, some FOXM1 was clearly retained on mitotic chromosomes with reticular shapes (Figure 3F, lower panels), suggesting some roles on double-stranded DNA even at the M phase. Collectively, these data suggested that FOXM1 could regulate the cell cycle and its inhibition caused G2/M arrest, indicating that FOXM1 could be a molecular target to inhibit cell proliferation, especially in AFP-positive HCC.

### 2.4. Carfilzomib Suppresses FOXM1 and Cell Proliferation in AFP-Positive HCC Cell Lines In Vitro

We evaluated FOXM1 expression in Huh7 cells and HCC cells derived from three patients (MT cells as AFP-positive HCC, and Kami41 and KM cells as AFP-negative HCC). We utilized our original patient-derived Kami41 and KM cells as the AFP-negative HCC cells because the available AFP-negative HCC cell lines such as HLE, HLF, and SK-Hep-1 cells have gene expression signatures that are characteristic of mesenchymal cells without hepatocyte-like gene expression patterns [16], and therefore did not match the AFP-negative HCC with good prognosis observed clinically. FOXM1 gene and protein expression was elevated in Huh7 and MT cells compared with Kami41 and KM cells (Figure 4A,B). For the pharmacological inhibition of FOXM1 in HCC, we searched the literature and found that siomycin A and thiostrepton, which are proteasome inhibitors, suppress FOXM1 protein expression [17,18]. Although the mechanism by which proteasome inhibitors reduce FOXM1 protein levels is unclear, bortezomib, a first-generation proteasome inhibitor approved for the treatment of multiple myeloma, is reported to suppresses FOXM1 mRNA and protein [17]. Accordingly, we tested the effect of carfilzomib, a strong, second-generation proteasome inhibitor that irreversibly binds to the 26S proteasome 19, on HCC cells. Interestingly, the treatment of AFP-positive Huh7 and MT cells with very low doses of carfilzomib (20 and 40 nM) for 24 h strongly reduced FOXM1 protein expression (Figure 4C). In contrast, the same treatment had no effect on FOXM1 protein expression in AFP-negative Kami41 and KM cells. Carfilzomib treatment at 20 nM for 24 h suppressed the viability of AFP-positive Huh7 and MT cells, but not AFP-negative Kami41 and KM cells (Figure 4D, *p* < 0.05). These data suggested that the proteasome inhibitor carfilzomib effectively suppressed cell proliferation in AFP-positive FOXM1-high HCC.

### 2.5. Carfilzomib Inhibits HCC Progression In Vivo

To evaluate the effect of carfilzomib on the progression of HCC, we utilized NOD/SCID mice for subcutaneous tumor xenotransplantation with Huh7 cells. When the tumor volume reached approximately 1000 mm^3^, these mice were defined as advanced HCC. We planned to evaluate the effect of DC101 (anti-mouse VEGFR2 antibody) on AFP-positive Huh7 cell growth given that recent studies clearly showed the efficacy of ramucirumab, an anti-human VEGFR2 monoclonal antibody, for the treatment of AFP-positive advanced HCC in humans [8,9]. We treated these mice with control IgG and vehicle, control IgG and carfilzomib, DC101 and vehicle, or DC101 and carfilzomib, according to the indicated schedule for 2 weeks (Figure 5A). We evaluated survival (*n* = 7 in each group) and tumor volume (*n* = 4 in each group) separately. In this condition, the combination of carfilzomib and DC101 prolonged the survival of mice compared with the control (*p* = 0.0072) (Figure 5A). Carfilzomib alone or DC101 alone also prolonged the survival compared with the control, with borderline significance (carfilzomib; *p* = 0.1, DC101; *p* = 0.07). The combination of carfilzomib and DC101 tended to prolong the survival of mice compared with single agents, but the difference did not reach statistical significance (carfilzomib; *p* = 0.14, DC101; *p* = 0.14). Besides, although carfilzomib treatment alone or DC101 treatment alone reduced tumor volume and weight, their combination suppressed tumor growth and weight in vivo (volumes; *p* = 0.015, weight; *p* = 0.063, by one-way ANOVA test) (Figure 5B–D). Interestingly, we found that only the combination of carfilzomib and DC101 treatment could reduce the tumor volumes compared with the control or carfilzomib/DC101 monotherapy (Figure 5C). Furthermore, carfilzomib treatment alone, DC101 treatment alone, and their combination reduced the number of AFP-positive, FOXM1-positive, or Ki-67-positive cells (Figure 5E,F, one-way ANOVA test). Western blot data confirmed the effect of carfilzomib, DC101, and their combination treatment on FOXM1 reduction (Figure 5G). Taken together, these data suggested the potential utility of treatment targeting microenvironmental VEGFR2 signaling with the blockade of FOXM1 for a better survival outcome in patients diagnosed with advanced-stage AFP-positive HCC.

## 3. Discussion

AFP-positive HCC shows a poor prognosis with a highly proliferative/invasive nature [19]. Previously, we reported that the combination of gadolinium ethoxybenzyl diethylenetriamine penta-acetic acid-enhanced magnetic resonance imaging and serum AFP can stratify HCC according to its stem/maturation status [15]. The transcription factor analysis identified FOXM1 as a candidate transcription factor controlling stemness in AFP-positive HCC [15]. Here, we found that FOXM1 indeed regulated AFP expression and cell proliferation in AFP-positive HCC. We also revealed that FOXM1 can be therapeutically abrogated by carfilzomib, an epoxyketone-based irreversible proteasome inhibitor. We proved that combination therapy with carfilzomib and DC101, rat anti-mouse VEGFR2-neutralizing antibody used as surrogate ramucirumab, suppressed the progression of AFP-positive HCC. DC101 or carfilzomib alone could suppress the expression of FOXM1 in Huh7 cells, and their combination further suppressed the expression of FOXM1, although the combination effects on FOXM1 were relatively mild, which might have resulted from the experimental schedule used in this study in vivo. Nevertheless, our data suggested that advanced AFP-positive HCC patients may receive a survival benefit by the addition of a reagent targeting FOXM1, such as carfilzomib, in combination with anti-VEGFR2 antibody treatment.

FOXM1 is a transcription factor that regulates cell proliferation, cell cycle progression, cell differentiation, DNA damage repair, tissue homeostasis, angiogenesis, and apoptosis [10,11,20]. Indeed, previous studies have reported that the increased expression of FOXM1 affects tumor growth and drug resistance in various solid tumors [21,22,23,24]. In this study, we clarified that FOXM1 was retained by dispersed chromatin at the M phase, when the bulk of DNA-binding proteins are excluded from condensed chromosomes [25], as analyzed by super-resolution microscopy image analysis. We further found that FOXM1 knockdown suppressed cell proliferation and induced G2/M cell cycle arrest in AFP-positive HCC cells, suggesting a role for FOXM1 at the M phase. Mechanistically, several reports have indicated that FOXM1 is required for central mitotic maturation by activating the Aurora kinase pathway or the cyclin B1/Cdc25 pathway [13]. Therefore, Aurora kinases or cyclin-dependent kinases could be additional molecular targets in AFP-positive FOXM1-high HCC, warranting future preclinical studies. Our data also showed that FOXM1-high HCC exhibited reduced expression of *SLCO1B1* and *CYP3A4*, which play a crucial role in mature hepatocyte metabolism. Because FOXM1 maintains cancer stem cells by inducing the epithelial–mesenchymal transition [22], a reduction in these mature hepatocyte metabolism-related genes might be accompanied by the loss of mature hepatocyte cell features due to the epithelial–mesenchymal transition.

The ubiquitin proteasome system plays a crucial role in controlling protein degradation to maintain the quality and quantity of various proteins [26]. Although the detailed mechanism is still under debate, high proteasome activity is noted in various types of cancer, including HCC. Potentially, high proteasome activity might result from genomic instability, the persistence of stressful conditions in the tumor microenvironment, and age-related proteasome imbalance [27]. Given that cell cycle control is tightly regulated by cyclins and cyclin-dependent kinases, which are degraded by the ubiquitin proteasome system, high proteasome activity in cancer cells might also reflect strong mitotic capacity in cancer [28]. Interestingly, according to recent evidence, proteasome inhibitors abrogate FOXM1 function in cancer [17,18].

Bortezomib is a first-generation proteasome inhibitor that reversibly and slowly inhibits the 26S proteasome, whereas carfilzomib irreversibly and strongly inhibits it [29,30,31]. Indeed, we found that treatment with 20 nM carfilzomib was sufficient to suppress the growth of FOXM1-positive HCC cells. Previously, we demonstrated that AFP is not just a tumor marker but also reflects liver cancer stem cell features with a poor survival outcome [1,6,7,15]. Considering the role of FOXM1 in proliferation and AFP expression in AFP-positive HCC cells, it is possible that the inhibition of FOXM1 with a very low concentration of carfilzomib might be therapeutically effective to suppress the proliferation of AFP-positive liver cancer stem cells. A recent study suggested the role of AFP in dendritic cell function through fatty-acid metabolism and oxidative phosphorylation, thus facilitating immune suppression [32]. Therefore, a reduction in AFP might be effective to activate immune cell function. Recently, chimeric antigen receptor T-cell therapy targeting a cancer stem cell marker Glypican 3 was tested for evaluating the safety profile in advanced HCC [33]. However, thus far, no clinically available treatment options have been utilized to target liver cancer stem cells directly; therefore, carfilzomib could be the first compound to target these cells and prolong overall survival.

Taken together, the results of this study demonstrated that (i) FOXM1 expression was high in AFP-positive HCC and (ii) the proteasome inhibitor carfilzomib suppressed FOXM1 expression and showed antitumor effects on AFP-positive HCC. Targeting FOXM1 in combination with VEGFR2 could be a novel therapeutic option to improve the survival of AFP-positive HCC patients.

## 4. Materials and Methods

### 4.1. Clinical Samples

One-hundred-thirty-three patients underwent surgical resection of HCC at Kanazawa University Hospital from 2008 to 2014 (Cohort 1). HCC and adjacent non-tumor tissues were fixed with formalin and used for immunohistochemical analysis. All patients provided written informed consent, and all tissue-acquisition procedures were approved by the Ethics Committee of Kanazawa University. Two-hundred-thirty-eight patients underwent surgical resection of HCC at the Liver Cancer Institute of Fudan University (Cohort 2). Portal vein invasion status was microscopically evaluated after surgery (Cohort 1) or macroscopically evaluated at the time of surgery (Cohort 2). Array data of Cohort 2 were publicly available (Gene Expression Omnibus accession number GSE14520).

### 4.2. RNA Interference

Small interfering RNAs (siRNAs) specifically targeting FOXM1 (s5249 and s5250) or the negative control (12,935,200) were purchased from Thermo Fisher Scientific (Waltham, MA, USA). A total of 2.0 × 10^5^ cells were seeded in 6-well plates 24 h before transfection. Transfection was performed using Lipofectamine RNAiMAX Transfection Reagent (Thermo Fisher Scientific) according to the manufacturer’s instructions. A concentration of 40 nM siRNA was used for each transfection.

### 4.3. Cell Culture and Reagents

Huh7 cells were obtained from the Japanese Collection of Research Bioresources Cell Bank (Saka, Japan) and authenticated by DNA profiling. Patient-derived cancer cells (MT, KM, and Kami41 cells) were established from resected HCC specimens at Kanazawa University Hospital as described previously. The cells were maintained in Dulbecco’s modified Eagle’s medium (Gibco, Grand Island, NY, USA) supplemented with 10% fetal bovine serum (Gibco) at 37 °C. Carfilzomib was obtained from Ono Pharmaceutical Co., Ltd. (Osaka, Japan). DC101 (anti-mouse VEGFR2 monoclonal antibody) was kindly provided by Eli Lilly and Company (Indianapolis, IN, USA).

### 4.4. Quantitative Reverse Transcription PCR

Total RNA was extracted using ISOSPIN Cell & Tissue RNA (Nippon Gene Co., Ltd., Tokyo, Japan) according to the manufacturer’s instructions. Quantitative PCR probes for FOXM1 (Hs1073586_m1) and AFP (Hs00173490_m1) were purchased from Applied Biosystems (Foster City, CA, USA). The expression of selected genes was determined in triplicate using the 7900 Sequence Detection System (Applied Biosystems). Each sample was normalized relative to the expression of a reference gene (18S rRNA). Quantitation of genes expressed in cell lines relative to Huh7 cells was performed using the ΔΔCT method.

### 4.5. Western Blotting

Whole-cell lysates were prepared using a radioimmunoprecipitation assay buffer with cOmplete, Mini, EDTA-free Protease Inhibitor Cocktail Tablets (Sigma-Aldrich Japan, Tokyo, Japan) and PhosSTOP EASYpack (Sigma-Aldrich, Tokyo, Japan). Anti-FOXM1 monoclonal (1:2000; ab207298; Abcam, Cambridge, MA, USA) and anti-β-actin monoclonal (#4967; Cell Signaling Technology, Danvers, MA, USA) antibodies were used. Immune complexes were visualized using Amersham ECL Prime Western Blotting Detection Reagents (RPN2232; GE Healthcare, Little Chalfont, UK) according to the manufacturer’s instructions.

### 4.6. Immunohistochemistry

Formalin-fixed, paraffin-embedded tissue sections (4 μm thick) were deparaffinized and hydrated. Following heat-induced epitope retrieval (citrate buffer, pH 6) and protein blocking (Protein Block Serum-Free; Dako, Carpinteria, CA, USA), the tissue sections were incubated with anti-FOXM1 (1:500; ab207298; Abcam), anti-AFP (1:100; c-19 sc-8108; Santa Cruz, Santa Cruz, CA, USA), or anti–Ki67 (1:100; ab16667; Abcam) antibodies overnight at 4 °C. After 3 washes with phosphate-buffered saline, the tissue sections were visualized using an EnVision+ Kit (Dako) according to the manufacturer’s instructions. The number of nuclei stained with anti-FOXM1 antibodies was calculated in each clinical HCC sample and defined as FOXM1-negative (<5% of nuclei stained), -low (5–25%), -moderate (25–50%), or -high (>50%), respectively. Approximately 40% and 60% of HCC cases were regarded as FOXM1-high (including FOXM1-moderate and -high) or FOXM1-low (including FOXM1-negative and -low), respectively.

### 4.7. Immunofluorescence and Super-Resolution Microscopy

Immunofluorescence analysis was performed using Huh7 cells, anti-FOXM1 antibody (Abcam), Alexa Fluor 488-conjugated anti-rabbit IgG secondary antibody (Thermo Fisher Scientific), and DAPI. Super-resolution images were obtained using the Dragonfly Confocal Imaging Platform (Andor, Belfast, UK).

### 4.8. Microarray Analysis

Affymetrix U133A2.0 gene expression data were obtained from the National Center for Biotechnology Information Gene Expression Omnibus database (accession number GSE14520). Forty percent of HCC samples were regarded as FOXM1-high according to the signal intensity data of FOXM1 and immunohistochemistry data. Pathway analysis was performed using MetaCore software (http://portal.genego.com (accessed on 26 May 2020)).

### 4.9. Cell Proliferation/Viability Assay

Briefly, 2.0 × 10^3^ cells (for cell proliferation) or 5.0 × 10^3^ cells (for cell viability) were seeded in 100 μL culture medium in each well of 96-well plates and pre-cultured in a CO_2_ incubator overnight. The cells were then transfected with siRNAs (for cell proliferation) or exposed to culture medium containing carfilzomib at the indicated concentration for 24 h (for cell viability). Cell proliferation/viability was evaluated using a Cell Counting Kit-8 (Dojindo Laboratories, Kumamoto, Japan) according to the manufacturer’s instruction.

### 4.10. Cell Cycle Analysis

Briefly, 2.0 × 10^5^ cells were seeded in 2 mL culture media and cultured overnight. SiRNAs targeting control or FOXM1 were transfected as described above. The single-cell suspension was prepared by trypsinization 48 h after transfection, and pre-chilled 70% ethanol was added to fix cells for 30 min on ice. Propidium iodide staining was performed using Cell Cycle Phase Determination Kit (Cayman Chemical, Ann Arbor, MI, USA), according to the manufacturer’s protocol. Cell cycle data was obtained using a FACSCalibur flowcytometer and analyzed by FlowJo Software v10.8.0 (Becton, Dickinson and Company, Franklin Lakes, NJ, USA).

### 4.11. Animal Studies

The study protocol was approved by the Kanazawa University Animal Care and Use Committee, and all procedures were performed in accordance with the guidelines and regulations of Kanazawa University. Six-week-old male non-obese diabetic (NOD)/severe combined immunodeficiency (SCID) mice were purchased from Charles River Laboratories, Inc. (Wilmington, MA, USA). Huh7 cells (1.0 × 10^6^ cells) were suspended in 200 µL Dulbecco’s modified Eagle’s medium and Matrigel (1:1) and subcutaneously injected into the flank region of the mice. The mice were randomly divided into four groups and treatment was started when tumor volume reached approximately 1000 mm^3^. The control vehicle (200 μL/day) or carfilzomib (6 mg/kg/day) was injected intraperitoneally on days 1, 2, 8, and 9. Control IgG (40 mg/kg) or anti-mouse VEGFR2 monoclonal antibodies (DC101, 40 mg/kg) were injected intraperitoneally on days 1, 4, 8, and 11. For survival analysis, the survival status of the mice (*n* = 7 in each group) were recorded every 2–3 days, and the mice were classified as “dead” and euthanized when tumor volume exceeded approximately 4000 mm^3^. For anti-tumor response analysis, the size of subcutaneous tumors (*n* = 4 in each group) was recorded every 2–3 days during treatment for 2 weeks. The mice were euthanized on day 14, and tumor volume and weight were measured. Subcutaneous tumors were excised and subsequently fixed in formalin and used for immunohistochemistry.

### 4.12. Statistical Analysis

Student’s *t*-test, chi-square test, Fisher’s exact test, Mann–Whitney test, one-way ANOVA test, and log-rank test were performed using GraphPad Prism 9.1 (GraphPad Software, San Diego, CA, USA). A *p*-value less than 0.05 was considered significant. All error bars in the figures represented standard error of the mean (SEM).

## Figures and Tables

**Figure 1 ijms-23-08305-f001:**
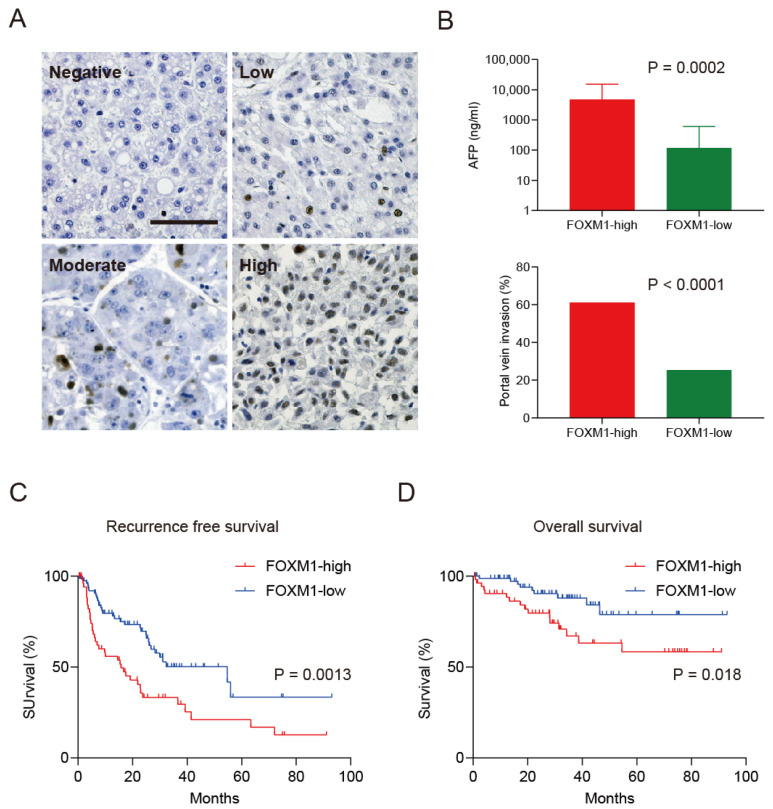
High FOXM1 expression is associated with poor prognosis in HCC patients (Cohort 1). (**A**) Representative immunohistochemistry images of FOXM1-negative (left upper panel), -low (right upper), -moderate (left lower), and -high HCC in surgically resected HCC tissue specimens (scale bar = 200 μm). (**B**) Serum AFP levels (upper panel) and frequency of microscopic portal vein invasion (lower panel) were higher in FOXM1-high HCC compared with FOXM1-low HCC (*p* = 0.0002, Mann–Whitney test for AFP, means ± SEM). (**C**) Kaplan–Meier survival analysis with the log-rank test of recurrence-free survival in FOXM1-high and -low HCC. Red, FOXM1-high (*n* = 54); blue, FOXM1-low (*n* = 79). (**D**) Kaplan–Meier survival analysis with the log-rank test of overall survival in FOXM1-high and -low HCC. Red, FOXM1-high (*n* = 54); blue, FOXM1-low (*n* = 79).

**Figure 2 ijms-23-08305-f002:**
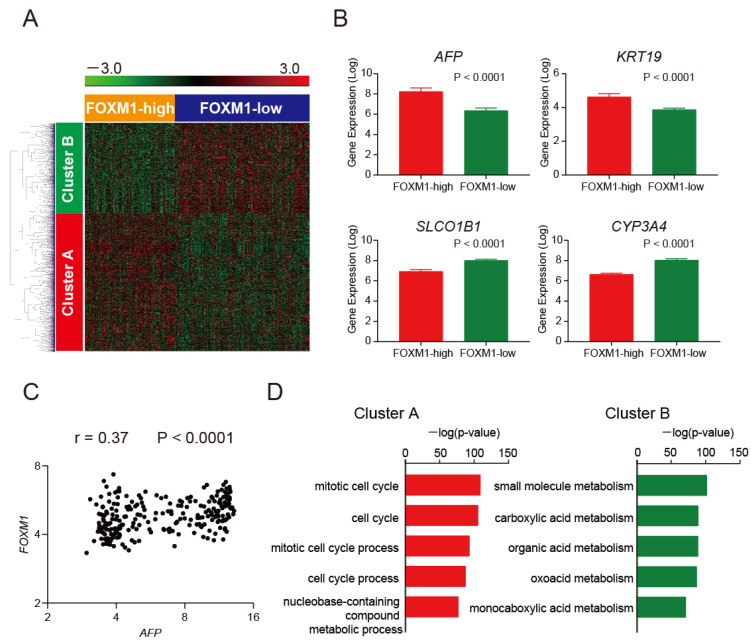
Transcriptomic characteristics of FOXM1-high and -low HCC (Cohort 2). (**A**) Hierarchical cluster analysis of 2119 genes differentially expressed between FOXM1-high and -low HCC. A total of 1275 genes and 844 genes were upregulated or downregulated, respectively, in FOXM1-high HCC compared with -low HCC (*p* < 0.001). (**B**) Signal intensity of probes corresponding to *AFP*, *KRT19*, *SLCO1B1*, and *CYP3A4* in FOXM1-high (red bar) and -low HCC (green bar) (Mann–Whitney test, means ± SEM). (**C**) Scatter plots analysis of AFP and FOXM1 expression in HCC (Spearman’s rank correlation coefficient). (**D**) Pathway analysis of FOXM1 co-regulated genes. Mitotic cell cycle processes were activated in FOXM1-high HCC (cluster A, red bars), whereas mature hepatocyte metabolism processes were inactivated in FOXM1-low HCC (cluster B, green bars). Significant processes are shown. (**E**) Kaplan–Meier survival analysis with the log-rank test of recurrence-free survival in FOXM1-high and -low HCC. Red, FOXM1-high (*n* = 94); blue, FOXM1-low (*n* = 143). (**F**) Kaplan–Meier survival analysis with the log-rank test of overall survival in FOXM1-high and -low HCC. Red, FOXM1-high (*n* = 94); blue, FOXM1-low (*n* = 143).

**Figure 3 ijms-23-08305-f003:**
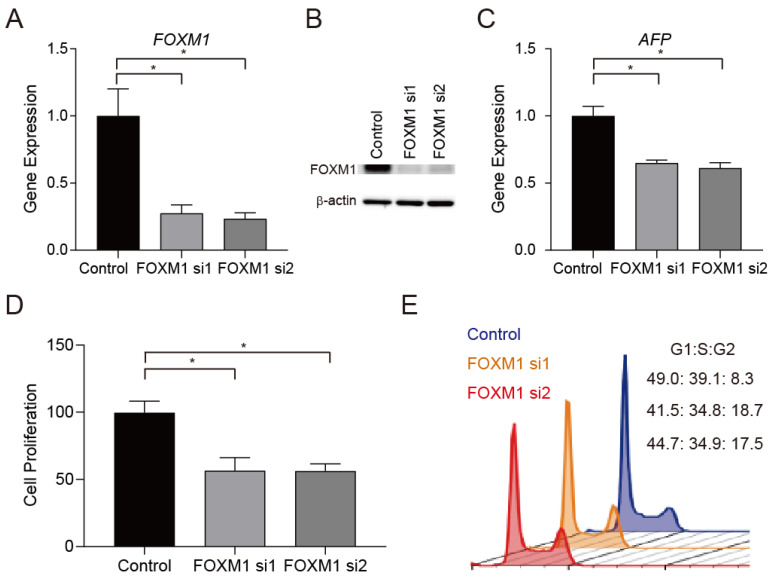
FOXM1 inhibition represses cell proliferation and AFP expression. (**A**) Knockdown of FOXM1 gene expression using siRNAs 48 h after transfection (FOXM1 si1 and si2) in Huh7 cells (*n* = 3 in each group, means ± SEM, Student’s *t*-tests, * *p* < 0.05). Experiments were performed twice. (**B**) Western blots of FOXM1 and β-actin in Huh7 cells treated with siRNAs (72 h after transfection). (**C**) Knockdown of FOXM1 suppressed AFP gene expression in Huh7 cells using siRNAs 48 h after transfection (*n* = 3 in each group, means ± SEM, Student’s *t*-tests, * *p* < 0.05). Experiments were performed twice. (**D**) FOXM1 knockdown inhibited cell proliferation (72 h after transfection) (*n* = 3 in each group, means ± SEM, Student’s *t*-tests, * *p* < 0.05). Experiments were performed twice. (**E**) The effect of FOXM1 knockdown on the cell cycle 48 h after transfection (blue, control siRNA; orange, FOXM1 si1; red, FOXM1 si2) Experiments were performed twice. (**F**) Super-resolution image analysis of FOXM1 in mitotic and non-mitotic Huh7 cells (scale bar = 10 μm).

**Figure 4 ijms-23-08305-f004:**
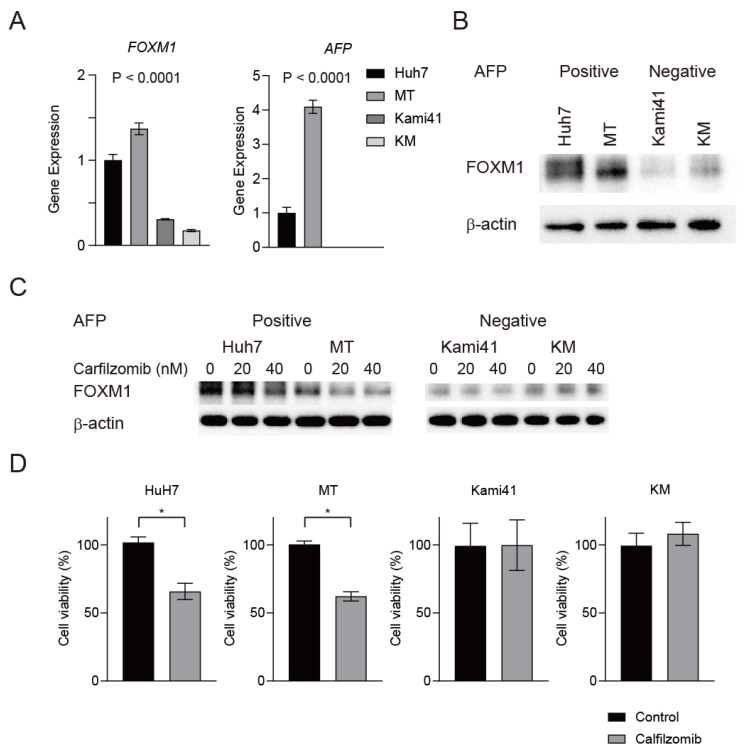
Carfilzomib inhibits FOXM1 protein expression in AFP-positive cells. (**A**) FOXM1 and AFP expression in Huh7, MT, Kami41, and KM cells (*n* = 3 in each group, means ± SEM, one-way ANOVA test). (**B**) Western blot analysis of FOXM1 in AFP-positive and -negative cells. (**C**) Western blot analysis of FOXM1 in AFP-positive and -negative cells treated with carfilzomib for 24 h. Experiments were performed twice. (**D**) Viability of Huh7, MT, Kami41, and KM cells treated with carfilzomib for 24 h (*n* = 3 in each group, means ± SEM, Student’s *t*-tests, * *p* < 0.05). Experiments were performed twice.

**Figure 5 ijms-23-08305-f005:**
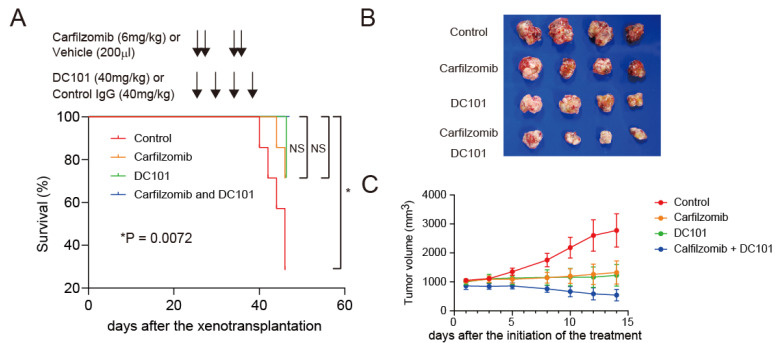
Carfilzomib combined with an anti-VEGFR2 antibody inhibits HCC progression. (**A**) Kaplan–Meier survival analysis with the log-rank test of NOD/SCID mice subcutaneously xenotransplanted with Huh7 cells. The mice were treated with control IgG and vehicle (red), control IgG and carfilzomib (orange), anti-VEGFR2 antibody and vehicle (green), or anti-VEGFR2 antibody and carfilzomib (blue) (*n* = 7 in each group). The control vehicle (200 μL/day) or carfilzomib (6 mg/kg/day) was injected intraperitoneally on days 1, 2, 8, and 9, as indicated by upper black arrows. Control IgG (40 mg/kg) or anti-mouse VEGFR2 monoclonal antibodies (DC101, 40 mg/kg) were injected intraperitoneally on days 1, 4, 8, and 11, as indicated by lower black arrows. (**B**) Photomicrographs of subcutaneous tumors that developed in NOD/SCID mice treated with the indicated reagents (*n* = 4 in each group). Mice were euthanized on day 14. Comparison of tumors with phosphate-buffered saline (control), carfilzomib (6 mg/kg), DC101 (40 mg/kg), or combined treatment with carfilzomib and DC101 for 2 weeks. (**C**) Subcutaneous tumor volume curves of Huh7 cells treated with the indicated reagents (*n* = 4 in each group, means ± SEM). (**D**) Subcutaneous tumor volume (upper panel) and weight (lower panel) in NOD/SCID mice treated with the indicated reagents on day 14 (one-way ANOVA test). (**E**) Representative immunohistochemistry staining images of FOXM1, AFP, and Ki-67 in subcutaneous tumors from NOD/SCID mice treated with the indicated reagents (scale bar = 100 μm). (**F**) Percentages of AFP-positive (left panel), FOXM1-positive (middle panel), and Ki-67-positive cells (right panel) counted three times from four independent immunohistochemistry staining images of subcutaneous tumors from NOD/SCID mice treated with the indicated reagents (means ± SEM, one-way ANOVA test). (**G**) Western blot of FOXM1 and β-actin in subcutaneous tumors from NOD/SCID mice treated with the indicated reagents.

**Table 1 ijms-23-08305-t001:** Clinical characteristics of HCC patients in Cohorts 1 and 2.

Parameter	Cohort 1 (*n* = 133)	Cohort 2 (*n* = 238)
Age (years, mean, SEM)	65.2, 0.9	50.7, 0.71
Sex (M/F)	99/34	208/30
AFP (ng/mL, median, 25–75%)	14, 10–198	184.4, 13.7–1629
BCLC stage (0-A, B-C, NA)	85/48/0	170/52/16
Virus (HBV/HCV/B + C/NBNC)	33/50/2/48	238/0/0/0
LC (yes/no)	71/62	219/19

AFP; alpha-fetoprotein, BCLC; Barcelona clinic liver cancer, HBV; hepatitis B virus, HCV; hepatitis C virus, NBNB; non-B non-C, LC; liver cirrhosis, NA; not available.

**Table 2 ijms-23-08305-t002:** Clinical characteristics of FOXM1-high and -low HCC patients in Cohort 1 and 2.

Cohort 1			
Parameter	FOXM1-High (*n* = 54)	FOXM1-Low (*n* = 79)	*p*-Value *
Age (years, mean, SEM)	66.5, 0.9	64.2, 1.2	0.33
Sex (M/F)	39/15	60/19	0.69
AFP (ng/mL, median, 25–75%)	182.5, 10–3075	10, 10–34	0.0002
Histologic grade ^†^ (I–II, II–III, III–IV) ^†^	13/24/17	29/40/10	0.024
Tumor size (mm, mean, SEM)	51.0, 5.8	38.1, 3.1	0.034
Microscopic PV invasion (yes/no)	33/21	24/55	0.0006
BCLC stage (0-A, B-C)	30/24	55/24	0.14
Virus (HBV/HCV/B + C/NBNC)	17/20/1/16	16/30/1/32	0.43
LC (yes/no)	32/22	39/40	0.29
**Cohort 2**			
**Parameter**	**FOXM1-High (*n* = 95)**	**FOXM1-Low (*n* = 143)**	** *p* ** **-Value ***
Age (years, mean, SEM)	50.6, 1.1	50.9, 0.9	0.5
Sex (M/F)	85/10	123/20	0.55
AFP (ng/mL, median, 25–75%)	589.4, 27.1–2908	98.9, 7.1–1210	0.0002
Macroscopic tumor thrombosis (yes/no)	20/75	11/132	0.0005
BCLC stage (0-A, B-C, NA)	57/28/10	113/24/6	0.005
LC (yes/no)	90/5	129/14	0.23

AFP; alpha-fetoprotein, BCLC; Barcelona clinic liver cancer, HBV; hepatitis B virus, HCV; hepatitis C virus, NBNB; non-B non-C, LC; liver cirrhosis, NA; not available, PV; portal vein. * Mann–Whitney test (age, AFP, and tumor size), Fisher’s exact test (LC), or *χ*^2^ test (sex, histologic grade, microscopic PV invasion, BCLC stage, and virus). ^†^ Edmondson–Steiner.

## Data Availability

Gene expression data were available at the National Center for Biotechnology Information Gene Expression Omnibus database (accession number GSE14520). Other data that support the findings of our study are available upon request from the corresponding author.

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
