# Peer review of "FOXM1 Is a Novel Molecular Target of AFP-Positive Hepatocellular Carcinoma Abrogated by Proteasome Inhibition"

_ijms, 2022, doi:10.3390/ijms23158305_

Round 1

Reviewer 1 Report

In the manuscript titled “FOXM1 is a novel molecular target of AFP-positive hepatocellular carcinoma abrogated by proteasome inhibition”, Li et al elucidate the association between FOXM1 expression and serum AFP levels in HCC patients. Further, they demonstrate that knockdown of FOXM1 decreases AFP expression and induces cell cycle arrest. They also identify an inhibitory effect of a proteasome inhibitor carfilzomib on FOXM1 protein expression. This manuscript makes important findings that have clinical relevance for HCC. The manuscript will be greatly strengthened with minor revisions, including indicating statistical analysis used for each assay, and the need to re-phrase the conclusion about the combination therapy. Please find below detailed comments.

Major comments:

·         The authors listed all the statistical assays used in the manuscript in the “Materials and Methods” section, but they did not specify the test used for each result. Please indicate the statistical assay used for each graph in the figure legends, including 1B, table 2 (which of the three tests was used for each parameter), 2B, 3A, 3B, 3C, 3D, 4D, 5A, 5C, and 5E.

·         Please include a Western blot showing decrease FOXM1 protein level with siRNA transfection (corresponding to figure 3A).

·         Please clarify if the decrease in AFP expression due to siRNA KD of FOXM1 is statistically significant or not (Fig. 3B). Also the effect on cell proliferation and number of cells in phases of cell cycle (Fig. 3C, 3D). Please add signs indicating statistical significance on the figures 3A, 3B, 3C, and 3D).

·         Lines 186 and 187 state that FOXM1 gene and protein expression were elevated in Huh7 and MT cells compared with Kami41 and KM cells. However, it is clear from Fig. 4A that FOXM1 gene expression in Huh7 is similar to Kami41 and KM cells, and is much less than Huh7 cells. Please clarify.

·         Please clarify how did survival of mice treated with carflizomib + DC101 compare with mice treated with single agent (Lines 221 and 222, figure 5A). Similarly, it is important to show the comparison between combination therapy vs. single therapy on tumor volume and tumor weight (Lines 224, 225). The same thing applies to AFP, FOXM1, and Ki67 measured in the 4 groups (Lines 225-228, Gif. 5E). An ANOVA test could be done and all the groups compared to evaluate if the combination therapy is better than single agent therapy.

·         The authors state that “We proved that combination therapy with carfilzomib and a VEGFR2-neutralizing antibody suppressed the progression of AFP-positive HCC more effectively than treatment with either agent alone.” However, they did not show by statistical analysis if the combination treatment is more effective than single drugs. This is a crucial point. If the combination has similar effect to single agent, the conclusion about the combination therapy needs to be re-phrased.

·         It would be very valuable to include a correlation curve demonstrating the correlation between serum AFP level and FOXM1 expression. While figures 1B and 2B proves that AFP is significantly higher in FOXM1-high HCC tissue as compared to FOXM1-low, the findings could be strengthened with a correlation curve including the full spectrum of FOXM1 expression levels.

·         It would be valuable to include Western blot analysis of FOXM1 in tumors treated with carfilzomib.

Minor comments:

·         Please define the abbreviations used in Tables 1 and 2 above or below the table, including LC, NBNC, HBV, HCV, BCLC, PV.

·         Is the standard error bar in the lower panel of Fig. 1B missing or too small?

·         Please correct Line 81: Table 2, not table 1. (The tumor size and histological grade are described in Table 2).

·         Please add to the figure legend of Figure 1 that the data is based on cohort 1 to make is easier for readers to differentiate between data in figures 1 and 2 relying on the figure legends. Also for figure 2.

·         In Materials and Methods statistical analysis section, please indicate what the error bars represent in all the figures, is it standard error or standard deviation?

·         A suggestion to replace the word “development” by “progression” in Line 211, because the tumor in this model does not initiate in vivo and the drug is administered after the tumor has already developed.

·         It would be valuable to add to the discussion a comparison between DC101 and ramucirumab.

·         It is not clear where is the blue line in Fig. 5A. Please clarify. Did these mice continue to survive beyond 50 days?

·         Please include information about cohort 2 patients under “4.1 Clinical Samples”.

·         Please explain how cell proliferation assays were performed in the “Materials and Methods” section. Is it similar to cell viability assays that include carfilzomib? Please clarify.

Author Response

In the manuscript titled “FOXM1 is a novel molecular target of AFP-positive hepatocellular carcinoma abrogated by proteasome inhibition”, Li et al elucidate the association between FOXM1 expression and serum AFP levels in HCC patients. Further, they demonstrate that knockdown of FOXM1 decreases AFP expression and induces cell cycle arrest. They also identify an inhibitory effect of a proteasome inhibitor carfilzomib on FOXM1 protein expression. This manuscript makes important findings that have clinical relevance for HCC. The manuscript will be greatly strengthened with minor revisions, including indicating statistical analysis used for each assay, and the need to re-phrase the conclusion about the combination therapy. Please find below detailed comments.

We would like to thank the reviewer for his/her professional positive comments to our study. Please find our point-by-point responses to the comments as described below.

Major comments:

  • The authors listed all the statistical assays used in the manuscript in the “Materials and Methods” section, but they did not specify the test used for each result. Please indicate the statistical assay used for each graph in the figure legends, including 1B, table 2 (which of the three tests was used for each parameter), 2B, 3A, 3B, 3C, 3D, 4D, 5A, 5C, and 5E.

We agree with the reviewer’s comments. We indicated the statistical assay used in the figure legends and revised the manuscript accordingly.

Page 3 line 94.

(B) Serum AFP levels (upper panel) and frequency of microscopic portal vein invasion (lower panel) were higher in FOXM1-high HCC compared with FOXM1-low HCC (P = 0.0002, Mann–Whitney test for AFP, means ± SEM). (C) Kaplan-Meier survival analysis with the log-rank test of recurrence-free survival in FOXM1-high and -low HCC. Red, FOXM1-high (n = 54); blue, FOXM1-low (n = 79). (D) Kaplan-Meier survival analysis with the log-rank test of overall survival in FOXM1-high and -low HCC. Red, FOXM1-high (n = 54); blue, FOXM1-low (n = 79).

Page 4 line 118.

*Mann–Whitney test (Age, AFP, and Tumor size), Fisher’s exact test (LC), or c2 test (Sex, Histo-logic grade, Microscopic PV invasion, BCLC stage, and Virus).

Page 5 line 147.

(B) Signal intensity of probes corresponding to AFP, KRT19, SLCO1B1, and CYP3A4 in FOXM1-high (red bar) and -low HCC (green bar) (Mann–Whitney test, means ± SEM).

Page 7 line 188.

(B) Western blots of FOXM1 and b-actin in Huh7 cells treated with siRNAs (72 h after transfection). (C) Knockdown of FOXM1 suppressed AFP gene expression in Huh7 cells using siRNAs 48 h after transfection (n = 3 in each group, means ± SEM, Student’s t-tests, * P < 0.05). Experiments were performed twice. (D) FOXM1 knockdown inhibited cell proliferation (72 h after transfection) (n = 3 in each group, means ± SEM, Student’s t-tests, * P < 0.05). Experiments were performed twice.

Page 9 line 229.

(D) Viability of Huh7, MT, Kami41, and KM cells treated with carfilzomib for 24 h (n = 3 in each group, means ± SEM, Student’s t-tests, * P < 0.05). Experiments were performed twice.

Page 11 line 265.

(A) Kaplan-Meier survival analysis with the log-rank test of NOD/SCID mice subcutaneously xenotransplanted with Huh7 cells. The mice were treated with control IgG and vehicle (red), control IgG and carfilzomib (orange), anti-VEGFR2 antibody and vehicle (green), or anti-VEGFR2 antibody and carfilzomib (blue) (n = 7 in each group). (B) Photomicrographs of subcutaneous tumors that developed in NOD/SCID mice treated with the indicated reagents (n = 4 in each group). Mice were euthanized on day 14. Comparison of tumors with phosphate-buffered saline (control), carfilzomib (6 mg/kg), DC101 (40 mg/kg), or combined treatment with carfilzomib and DC101 for 2 weeks. (C) Subcutaneous tumor volume curves of Huh7 cells treated with the indicated reagents (n = 4 in each group, means ± SEM). (D) Subcutaneous tumor volume (upper panel) and weight (lower panel) in NOD/SCID mice treated with the indicated reagents on day 14 (one-way ANOVA test). (E) Representative immunohistochemistry staining images of FOXM1, AFP, and Ki-67 in subcutaneous tumors from NOD/SCID mice treated with the indicated reagents (scale bar = 100 μm). (F) Percentages of AFP-positive (left panel), FOXM1-positive (middle panel), and Ki-67-positive cells (right panel) counted three times from four independent immunohistochemistry staining images of subcutaneous tumors from NOD/SCID mice treated with the indicated reagents (means ± SEM, one-way ANOVA test).

  • Please include a Western blot showing decrease FOXM1 protein level with siRNA transfection (corresponding to figure 3A).

We agree with the reviewer’s comment. We added the Western blot in new Fig. 3B. We revised the manuscript accordingly.

Page 6 line 163.

Western blot showed the reduction of FOXM1 protein using the same condition (Figure 3B).

  • Please clarify if the decrease in AFP expression due to siRNA KD of FOXM1 is statistically significant or not (Fig. 3B). Also the effect on cell proliferation and number of cells in phases of cell cycle (Fig. 3C, 3D). Please add signs indicating statistical significance on the figures 3A, 3B, 3C, and 3D).

We agree with the reviewer’s comments. We revised the Fig. 3A, new Fig. 3C and D, and add signs indicating statistical significance. For cell cycle analysis (new Fig. 3E), we could not perform statistical significance because we repeated the cell cycle experiment only twice which is generally acceptable for cell cycle analysis. We emphasized that we repeated the experiment twice to confirm the data reproducibility. We revised the manuscript accordingly.

Page 7 line 189.

(C) Knockdown of FOXM1 suppressed AFP gene expression in Huh7 cells using siRNAs 48 h after transfection (n = 3 in each group, means ± SEM, Student’s t-tests, * P < 0.05). Experiments were performed twice. (D) FOXM1 knockdown inhibited cell proliferation (72 h after transfection) (n = 3 in each group, means ± SEM, Student’s t-tests, * P < 0.05). Experiments were performed twice. (E) The effect of FOXM1 knockdown on the cell cycle 48 h after transfection (blue, control siRNA; orange, FOXM1 si1; red, FOXM1 si2) Experiments were performed twice.

  • Lines 186 and 187 state that FOXM1 gene and protein expression were elevated in Huh7 and MT cells compared with Kami41 and KM cells. However, it is clear from Fig. 4A that FOXM1 gene expression in Huh7 is similar to Kami41 and KM cells, and is much less than Huh7 cells. Please clarify.

We would like to thank the reviewer’s professional comments. We agreed the discrepancy of mRNA and protein data in FOXM1 expression in cell lines and we re-evaluated the expression status of FOXM1. We would like to apologize that the mRNA expression data was mis-calculated and wrong. We corrected the Fig. 4A accordingly.

  • Please clarify how did survival of mice treated with carflizomib + DC101 compare with mice treated with single agent (Lines 221 and 222, figure 5A). Similarly, it is important to show the comparison between combination therapy vs. single therapy on tumor volume and tumor weight (Lines 224, 225). The same thing applies to AFP, FOXM1, and Ki67 measured in the 4 groups (Lines 225-228, Gif. 5E). An ANOVA test could be done and all the groups compared to evaluate if the combination therapy is better than single agent therapy.

We agree with the reviewer’s comments. We performed Kaplan-Meier analysis with log-rank tests for two groups comparison (Control and carfilzomib/DC101). We clearly stated the analysis in the revised manuscript. During this revision, we noticed that number of mice used for survival analysis was not 5 but actually 7 in each group. We corrected the number oof mice accordingly. We also added the data to show the four-group comparison in terms of tumor volumes and sizes by one-way ANOVA tests, as suggested by the reviewer. We would like to thank the reviewer’s professional comments and revised the manuscript accordingly.

Page 9 line 242.

We evaluated survival (n = 7 in each group) and tumor volume (n = 4 in each group) separately. In this condition, the combination of carfilzomib and DC101 prolonged the survival of mice compared with control (P = 0.0072) (Figure 5A). Carfilzomib alone or DC101 alone also prolonged the survival compared with control with borderline significance (carfilzomib; P = 0.1, DC101; P = 0.07). The combination of carfilzomib and DC101 tended to prolong the survival of mice compared with single agents but the difference did not reach statistical significance (carfilzomib; P = 0.14, DC101; P = 0.14). Besides, although carfilzomib treatment alone or DC101 treatment alone reduced tumor volume and weight, their combination suppressed tumor growth and weight in vivo (volumes; P = 0.015, weight; P = 0.063, by one-way ANOVA test) (Figure 5B, C, and D). Interestingly, we found that only combination of carfilzomib and DC101 treatment could reduce the tumor volumes compared with control or carfilzomib/DC101 monotherapy (Figure 5C). Furthermore, carfilzomib treatment alone or DC101 treatment alone reduced the number of AFP-positive, FOXM1-positive, or Ki-67-positive cells, their combination further reduced the number of these marker-positive cells (Figure 5E and F, one-way ANOVA test).

  • The authors state that “We proved that combination therapy with carfilzomib and a VEGFR2-neutralizing antibody suppressed the progression of AFP-positive HCC more effectively than treatment with either agent alone.” However, they did not show by statistical analysis if the combination treatment is more effective than single drugs. This is a crucial point. If the combination has similar effect to single agent, the conclusion about the combination therapy needs to be re-phrased.

We agree with the reviewer’s comments. We confirm that combination therapy did not prolong the mice survival with statistical significance compared with monotherapy. We agree that the message was too strong and revised the manuscript accordingly.

Page 12 line 293.

We proved that combination therapy with carfilzomib and DC101, rat anti-mouse VEGFR2-neutralizing antibody used as surrogate ramucirumab, suppressed the progression of AFP-positive HCC. DC101 or carfilzomib alone could suppress the expression of FOXM1 in Huh7 cells, and their combination further suppressed the expression of FOXM1, although the combination effects on FOXM1 was relatively mild, which might be resulted from the experimental schedule used in this study in vivo.

  • It would be very valuable to include a correlation curve demonstrating the correlation between serum AFP level and FOXM1 expression. While figures 1B and 2B proves that AFP is significantly higher in FOXM1-high HCC tissue as compared to FOXM1-low, the findings could be strengthened with a correlation curve including the full spectrum of FOXM1 expression levels.

According to the reviewer’s suggestion, we added the scatter plots of FOXM1 and AFP using the microarray data in Fig. 2C and revised the manuscript accordingly.

Page 4 line 129.

Weak positive correlation was observed between FOXM1 and AFP signal intensities in 238 microarray samples (Figure 2C, r = 0.37, P < 0.0001, Spearman’s rank correlation co-efficient).

Page 5 line 149.

(C) Scatter plots analysis of AFP and FOXM1 expression in HCC (Spearman’s rank correlation coefficient).

  • It would be valuable to include Western blot analysis of FOXM1 in tumors treated with carfilzomib.

According to the reviewer’s suggestion, we added Western blot of FOXM1 in Fig. 5G and revised the manuscript accordingly.

Page 10 line 257.

Western blot data confirmed the effect of carfilzomib, DC101, and their combination treatment on FOXM1 reduction.

Page 12 line 282.

(G) Western blot of FOXM1 and b-actin in subcutaneous tumors from NOD/SCID mice treated with the indicated reagents.

Minor comments:

  • Please define the abbreviations used in Tables 1 and 2 above or below the table, including LC, NBNC, HBV, HCV, BCLC, PV.

We agree with the reviewer’s comment and revised Table 1 and 2 accordingly.

Page 2 line 77.

AFP; Alpha-fetoprotein, BCLC; Barcelona Clinic Liver Cancer, HBV; Hepatitis B Virus, HCV; Hepatitis C Virus, NBNB; non-B non-C, LC; Liver Cirrhosis, NA; Not Available.

Page 4 line 116.

AFP; Alpha-fetoprotein, BCLC; Barcelona Clinic Liver Cancer, HBV; Hepatitis B Virus, HCV; Hepatitis C Virus, NBNB; non-B non-C, LC; Liver Cirrhosis, NA; Not Available, PV; Portal Vein.

  • Is the standard error bar in the lower panel of Fig. 1B missing or too small?

This is because frequency of portal vein invasion in each group was single value. We agree that the data presentation might be confusing and revised the figure legends accordingly.

Page 3 line 94.

(B) Serum AFP levels (upper panel) and frequency of microscopic portal vein invasion (lower panel) were higher in FOXM1-high HCC compared with FOXM1-low HCC (P = 0.0002, Mann–Whitney test for AFP, means ± SEM).

  • Please correct Line 81: Table 2, not table 1. (The tumor size and histological grade are described in Table 2).

We would like to thank the comment and revised the manuscript accordingly.

Page 2 line 83.

We evaluated the clinicopathological characteristics of the FOXM1-high and -low HCC cases, and FOXM1-high HCC was significantly associated with high serum AFP levels, poorly differentiated histological findings, large tumor size, and high frequency of microscopic portal vein invasion (Figure 1B and Table 2).

  • Please add to the figure legend of Figure 1 that the data is based on cohort 1 to make is easier for readers to differentiate between data in figures 1 and 2 relying on the figure legends. Also for figure 2.

We agree with the reviewer’s comments and revised the figure legends accordingly.

Page 3 line 91.

Figure 1. High FOXM1 expression is associated with poor prognosis in HCC patients (Cohort 1).

Page 5 line 144.

Figure 2. Transcriptomic characteristics of FOXM1-high and -low HCC (Cohort 2).

  • In Materials and Methods statistical analysis section, please indicate what the error bars represent in all the figures, is it standard error or standard deviation?

We agree with the comments and revised the manuscript accordingly.

Page 15 line 468.

All error bars in the figures represented standard error of the mean (SEM).

  • A suggestion to replace the word “development” by “progression” in Line 211, because the tumor in this model does not initiate in vivo and the drug is administered after the tumor has already developed.

We revised the manuscript accordingly.

Page 9 line 233

2.5 Carfilzomib inhibits HCC progression in vivo

  • It would be valuable to add to the discussion a comparison between DC101 and ramucirumab.

We added the information of DC101 and ramucirumab in the discussion section.

Page 12 line 293.

We proved that combination therapy with carfilzomib and DC101, rat anti-mouse VEGFR2-neutralizing antibody used as surrogate ramucirumab, suppressed the progression of AFP-positive HCC.

  • It is not clear where is the blue line in Fig. 5A. Please clarify. Did these mice continue to survive beyond 50 days?

We agree with the reviewer’s comment and revised the Fig.5A. All these mice survived beyond 50 days.

  • Please include information about cohort 2 patients under “4.1 Clinical Samples”.

We added the cohort 2 information and revised the manuscript.

Page 13 line 358.

Two hundred thirty-eight patients underwent surgical resection of HCC at the Liver Cancer Institute of Fudan University (Cohort 2). Portal vein invasion status was microscopically evaluated after surgery (Cohort 1) or macroscopically evaluated at the time of surgery (Cohort 2). Array data of cohort 2 were publicly available (Gene Expression Omnibus accession number GSE14520).

  • Please explain how cell proliferation assays were performed in the “Materials and Methods” section. Is it similar to cell viability assays that include carfilzomib? Please clarify.

We apologize that we did not include the cell proliferation assay protocol. We revised the manuscript accordingly.

Page 14 line 427.

Briefly, 2.0 × 103 cells (for cell proliferation) or 5.0 × 103 cells (for cell viability) were seeded in 100 μL culture medium in each well of 96-well plates and pre-cultured in a CO2 incubator overnight. The cells were then transfected with siRNAs (for cell proliferation) or exposed to culture medium containing carfilzomib at the indicated concentration for 24 h (for cell viability). Cell proliferation/viability was evaluated using a Cell Counting Kit-8 (Dojindo Laboratories, Kumamoto, Japan) according to the manufacturer’s instruction.

Reviewer 2 Report

In this manuscript, authors have identified FOXM1 as a molecular target that is upregulated in alpha-fetoprotein (AFP)-positive hepatocellular carcinoma (HCC). Authors also show that FOXM1-high HCC have poor prognosis and a proteasome inhibitor, carfilzomib could attenuate FOXM1 expression and suppress proliferation of AFP-positive HCC cells. In addition, carfilzomib in combination with VEGFR2 blockade suppressed the growth of AFP-positive HCC tumors and enhanced survival of tumor-bearing mice. Since effective therapeutic options for AFP-positive HCC are limited, this study has important implications for treatment of patients with AFP-positive HCC. However, there are few suggestions:

1.    Figure 4, lines 186-187: Authors mention that both FOXM1 gene and protein expression was elevated in Huh7 and MT cells compared to Kami41 and KM cells. However, gene expression of FOXM1 is not different in Huh7 cells compared to Kami41 and KM cells although protein expression is higher. Also, it is important to mention here about the expression of AFP that remains high in Huh7 cells, suggesting that post-transcriptional regulation is important in Huh7 cells that maintains the high level of FOXM1 protein and in turn AFP.

2.    Figure 5: They are subcutaneous tumors; therefore, authors should show complete tumor growth profiles and these can be shown even from the same experiment where survival was monitored. This is also important since it appears that VEGFR2 blockade itself is inducing potent anti-tumor effects and therefore combining it with carfilzomib does not further enhance the effects. The statistical significance between controls and each of the monotherapies and also between monotherapies and the combination group should be shown to make logical conclusions. Similar is true for figure 5E since monotherapies appear to be as good as the combination treatment, making it difficult to understand why anti-tumor effects (tumor growth delay and survival) are better with the combination. This may suggest that dose or schedule of DC101 as used here may not be optimal to combine with carfilzomib. This also needs to be discussed in the discussion. Accordingly, the statement on lines 258-261, “We proved that combination therapy with carfilzomib and a VEGFR2-neutralizing antibody suppressed the progression of AFP-positive HCC more effectively than treatment with either agent alone.” is not true since comparisons between either agents alone and the combination group are not made and at least for VEGFR2-neutralizing antibody effects appears to be as good as with the combination.

3.    Discussion will be strengthened if other strategies that have been utilized/considered based on overexpression of AFP that influences immune suppression and stem cell properties in addition to angiogenesis (VEGFR2 blockade as used in the present study) are discussed; for example, use of other anti-cancer drugs such as doxorubicin or role of CAR T cell based therapies.

4.    Methods: Include for portal vein invasion shown in Fig. 1 B; cell cycle distribution studies shown in Fig. 3D; and samples collection for IHC in section 4.10.

In addition, minor concerns and suggestions are:

1.    Line 50: Change “mechanical” to “mechanistic”.

2.    Expand abbreviations such as HBV, HCV, BCLC, NBNC, LC, NA etc.

3.    Line 81: Table 2 should be referred with Figure 1B instead of Table 1.

4.    Line 111: †Edmondson–Steiner; the symbol is not shown in the table 2.

5.    It is important to provide some more experimental details (may be given in the figure legends also so it becomes easier to follow), especially number of times experiments were repeated (for all the figures) as well as time points when various analyses were performed, (ex for Figure 3E; tumor volumes and weights, IHC data etc.).

6.    Please provide statistical analysis (p-values) for Figures 3A-C and 4.

7.    Lines 216-218: Provide reference for…. recent studies clearly showed the efficacy of ramucirumab, an anti-human VEGFR2 monoclonal antibody, for the treatment of AFP-positive advanced HCC in humans.

Author Response

In this manuscript, authors have identified FOXM1 as a molecular target that is upregulated in alpha-fetoprotein (AFP)-positive hepatocellular carcinoma (HCC). Authors also show that FOXM1-high HCC have poor prognosis and a proteasome inhibitor, carfilzomib could attenuate FOXM1 expression and suppress proliferation of AFP-positive HCC cells. In addition, carfilzomib in combination with VEGFR2 blockade suppressed the growth of AFP-positive HCC tumors and enhanced survival of tumor-bearing mice. Since effective therapeutic options for AFP-positive HCC are limited, this study has important implications for treatment of patients with AFP-positive HCC. However, there are few suggestions:

We would like to thank the reviewer for his/her professional positive comments to our study. Please find our point-by-point responses to the comments as described below.

  1. Figure 4, lines 186-187: Authors mention that both FOXM1 gene and protein expression was elevated in Huh7 and MT cells compared to Kami41 and KM cells. However, gene expression of FOXM1 is not different in Huh7 cells compared to Kami41 and KM cells although protein expression is higher. Also, it is important to mention here about the expression of AFP that remains high in Huh7 cells, suggesting that post-transcriptional regulation is important in Huh7 cells that maintains the high level of FOXM1 protein and in turn AFP.

We completely agree with the reviewer’s professional comments. We also found the discrepancy of mRNA and protein data in FOXM1 expression in cell lines, and we re-evaluated the expression status of FOXM1 independently. We would like to apologize that the mRNA expression data was mis-calculated and wrong. We corrected the Figure 4A accordingly. We would like to thank this Reviewer again for pointing out the important point.

  1. Figure 5: They are subcutaneous tumors; therefore, authors should show complete tumor growth profiles and these can be shown even from the same experiment where survival was monitored. This is also important since it appears that VEGFR2 blockade itself is inducing potent anti-tumor effects and therefore combining it with carfilzomib does not further enhance the effects. The statistical significance between controls and each of the monotherapies and also between monotherapies and the combination group should be shown to make logical conclusions. Similar is true for figure 5E since monotherapies appear to be as good as the combination treatment, making it difficult to understand why anti-tumor effects (tumor growth delay and survival) are better with the combination. This may suggest that dose or schedule of DC101 as used here may not be optimal to combine with carfilzomib. This also needs to be discussed in the discussion. Accordingly, the statement on lines 258-261, “We proved that combination therapy with carfilzomib and a VEGFR2-neutralizing antibody suppressed the progression of AFP-positive HCC more effectively than treatment with either agent alone.” is not true since comparisons between either agents alone and the combination group are not made and at least for VEGFR2-neutralizing antibody effects appears to be as good as with the combination.

We respectfully agree with reviewer’s comments. Unfortunately, for mouse survival study, we could not reach the exact tumor size data especially after day 42 when control mouse started to die. Instead, we do have tumor growth curve data for tumor volume analysis. Interestingly, we found the mild reduction of tumor volumes when treated with DC101/calfilzomib combination treatment from baseline, whereas control or monotherapy treated groups showed rapid or mild increase of tumor volumes. We appreciate the reviewer’s comments and added these data in revised Figure 5C. We also re-calculated the mice survival data. We confirmed that combination therapy did not prolong the mice survival with statistical significance compared with monotherapy. We agree that the message was too strong and revised the manuscript accordingly. For tumor size/volume comparison, we re-analyzed the data using one-way ANOVA test for four-group comparison, as suggested by the other Reviewer. We revised the manuscript and added these points in the discussion section accordingly.

Page 9 line 240.

We treated these mice with control IgG and vehicle, control IgG and carfilzomib, DC101 and vehicle, or DC101 and carfilzomib, according to the indicated schedule for 2 weeks (Figure 5A). We evaluated survival (n = 7 in each group) and tumor volume (n = 4 in each group) separately. In this condition, the combination of carfilzomib and DC101 pro-longed the survival of mice compared with control (P = 0.0072) (Figure 5A). Carfilzomib alone or DC101 alone also prolonged the survival compared with control with border-line significance (carfilzomib; P = 0.1, DC101; P = 0.07). The combination of carfilzomib and DC101 tended to prolong the survival of mice compared with single agents but the difference did not reach statistical significance (carfilzomib; P = 0.14, DC101; P = 0.14). Besides, although carfilzomib treatment alone or DC101 treatment alone reduced tumor volume and weight, their combination suppressed tumor growth and weight in vivo (volumes; P = 0.015, weight; P = 0.063, by one-way ANOVA test) (Figure 5B, C, and D). Interestingly, we found that only combination of carfilzomib and DC101 treatment could reduce the tumor volumes compared with control or carfilzomib/DC101 monotherapy (Figure 5C). Furthermore, carfilzomib treatment alone or DC101 treatment alone reduced the number of AFP-positive, FOXM1-positive, or Ki-67-positive cells, their combination further reduced the number of these marker-positive cells (Figure 5E and F, one-way ANOVA test).

Page 11 line 265.

(A) Kaplan-Meier survival analysis with the log-rank test of NOD/SCID mice subcutaneously xenotransplanted with Huh7 cells. The mice were treated with control IgG and vehicle (red), control IgG and carfilzomib (orange), anti-VEGFR2 antibody and vehicle (green), or anti-VEGFR2 antibody and carfilzomib (blue) (n = 7 in each group). (B) Photomicrographs of subcutaneous tumors that developed in NOD/SCID mice treated with the indicated reagents (n = 4 in each group). Mice were euthanized on day 14. Comparison of tumors with phosphate-buffered saline (control), carfilzomib (6 mg/kg), DC101 (40 mg/kg), or combined treatment with carfilzomib and DC101 for 2 weeks. (C) Subcutaneous tumor volume curves of Huh7 cells treated with the indicated reagents (n = 4 in each group, means ± SEM). (D) Subcutaneous tumor volume (upper panel) and weight (lower panel) in NOD/SCID mice treated with the indicated reagents on day 14 (one-way ANOVA test). (E) Representative immunohistochemistry staining images of FOXM1, AFP, and Ki-67 in subcutaneous tumors from NOD/SCID mice treated with the indicated reagents (scale bar = 100 μm). (F) Percentages of AFP-positive (left panel), FOXM1-positive (middle panel), and Ki-67-positive cells (right panel) counted three times from four independent immunohistochemistry staining images of subcutaneous tumors from NOD/SCID mice treated with the indicated reagents (means ± SEM, one-way ANOVA test).

Page 12 line 293.

We proved that combination therapy with carfilzomib and DC101, rat anti-mouse VEGFR2-neutralizing antibody used as surrogate ramucirumab, suppressed the progression of AFP-positive HCC. DC101 or carfilzomib alone could suppress the expression of FOXM1 in Huh7 cells, and their combination further suppressed the expression of FOXM1, although the combination effects on FOXM1 was relatively mild, which might be resulted from the experimental schedule used in this study in vivo.

  1. Discussion will be strengthened if other strategies that have been utilized/considered based on overexpression of AFP that influences immune suppression and stem cell properties in addition to angiogenesis (VEGFR2 blockade as used in the present study) are discussed; for example, use of other anti-cancer drugs such as doxorubicin or role of CAR T cell based therapies.

We agree with the reviewer’s comments. We revised the discussion section accordingly.

Page 13 line 339.

A recent study suggested the role of AFP on dendritic cell function through fatty acid metabolism and oxidative phosphorylation, thus facilitating the immune suppression [32]. Therefore, reduction of AFP might be effective to activate immune cell function. Recently, chimeric antigen receptor T-cell therapy targeting a cancer stem cell marker Glypican 3 was tested for evaluating safety profile in advanced HCC [33].

  1. Methods: Include for portal vein invasion shown in Fig. 1 B; cell cycle distribution studies shown in Fig. 3D; and samples collection for IHC in section 4.10.

We agree with the reviewer’s comments and added the detailed methods in the Methods section.

Page 13 line 354.

One hundred thirty-three patients underwent surgical resection of HCC at Kanazawa University Hospital from 2008 to 2014 (Cohort 1). HCC and adjacent non-tumor tissues were fixed with formalin and used for immunohistochemical analysis. All patients provided written informed consent, and all tissue acquisition procedures were approved by the Ethics Committee of Kanazawa University. Two hundred thirty-eight patients underwent surgical resection of HCC at the Liver Cancer Institute of Fudan University (Cohort 2). Portal vein invasion status was microscopically evaluated after surgery (Cohort 1) or macroscopically evaluated at the time of surgery (Cohort 2). Array data of cohort 2 were publicly available (Gene Expression Omnibus accession number GSE14520).

Page 14 line 435

4.10 Cell cycle analysis

Briefly, 2.0 × 105 cells were seeded in 2 ml culture media and cultured overnight. Si-RNAs targeting control or FOXM1 were transfected as described above. Single cell suspension was prepared by trypsinization 48 h after transfection, and pre-chilled 70% ethanol was added to fix cells for 30 min on ice. Propidium iodide staining was per-formed using Cell Cycle Phase Determination Kit (Cayman Chemical, Ann Arbor, MI), according to the manufacturer’s protocol. Cell cycle data was obtained using a FACSCalibur flowcytometer and analyzed by FlowJo Software v10.8.0 (Becton, Dickinson and Company, Franklin Lakes, NJ).

Page 15 line 460.

The mice were euthanized on day 14, and tumor volume and weight were measured. Subcutaneous tumors were excised and subsequently fixed in formalin and used for immunohistochemistry.

In addition, minor concerns and suggestions are:

  1. Line 50: Change “mechanical” to “mechanistic”.

We revised the manuscript accordingly.

Page 2 line 52.

However, the mechanistic link between AFP production and VEGFR2 expression is unclear.

  1. Expand abbreviations such as HBV, HCV, BCLC, NBNC, LC, NA etc.

We revised the manuscript accordingly.

Page 2 line 77.

AFP; Alpha-fetoprotein, BCLC; Barcelona Clinic Liver Cancer, HBV; Hepatitis B Virus, HCV; Hepatitis C Virus, NBNB; non-B non-C, LC; Liver Cirrhosis, NA; Not Available.

Page 4 line 116.

AFP; Alpha-fetoprotein, BCLC; Barcelona Clinic Liver Cancer, HBV; Hepatitis B Virus, HCV; Hepatitis C Virus, NBNB; non-B non-C, LC; Liver Cirrhosis, NA; Not Available, PV; Portal Vein.

  1. Line 81: Table 2 should be referred with Figure 1B instead of Table 1.

We revised the manuscript accordingly.

Page 2 line 83.

We evaluated the clinicopathological characteristics of the FOXM1-high and -low HCC cases, and FOXM1-high HCC was significantly associated with high serum AFP levels, poorly differentiated histological findings, large tumor size, and high frequency of mi-croscopic portal vein invasion (Figure 1B and Table 2).

  1. Line 111: †Edmondson–Steiner; the symbol is not shown in the table 2.

We revised Table 2 accordingly.

  1. It is important to provide some more experimental details (may be given in the figure legends also so it becomes easier to follow), especially number of times experiments were repeated (for all the figures) as well as time points when various analyses were performed, (ex for Figure 3E; tumor volumes and weights, IHC data etc.).

We agree with the reviewer’s comments. We added the experimental details in the figure legends.

Page 7 line 185.

(A) Knockdown of FOXM1 gene expression using siRNAs 48 h after transfection (FOXM1 si1 and si2) in Huh7 cells (n = 3 in each group, means ± SEM, Student’s t-tests, * P < 0.05). Experiments were performed twice. (B) Western blots of FOXM1 and b-actin in Huh7 cells treated with siRNAs (72 h after transfection). (C) Knockdown of FOXM1 suppressed AFP gene expression in Huh7 cells using siRNAs 48 h after transfection (n = 3 in each group, means ± SEM, Student’s t-tests, * P < 0.05). Experiments were performed twice. (D) FOXM1 knockdown inhibited cell proliferation (72 h after transfection) (n = 3 in each group, means ± SEM, Student’s t-tests, * P < 0.05). Experiments were performed twice. (E) The effect of FOXM1 knockdown on the cell cycle 48 h after transfection (blue, control siRNA; orange, FOXM1 si1; red, FOXM1 si2) Experiments were performed twice.

Page 9 line 225

(A) FOXM1 and AFP expression in Huh7, MT, Kami41, and KM cells (n = 3 in each group, means ± SEM). (B) Western blot analysis of FOXM1 in AFP-positive and -negative cells. (C) Western blot analysis of FOXM1 in AFP-positive and -negative cells treated with carfilzomib for 24 h. Experiments were performed twice. (D) Viability of Huh7, MT, Kami41, and KM cells treated with carfilzomib for 24 h (n = 3 in each group, means ± SEM, Student’s t-tests, * P < 0.05). Experiments were performed twice.

Page 11 line 265.

(A) Kaplan-Meier survival analysis with the log-rank test of NOD/SCID mice subcutaneously xenotransplanted with Huh7 cells. The mice were treated with control IgG and vehicle (red), control IgG and carfilzomib (orange), anti-VEGFR2 antibody and vehicle (green), or anti-VEGFR2 antibody and carfilzomib (blue) (n = 7 in each group). (B) Photomicrographs of subcutaneous tumors that developed in NOD/SCID mice treated with the indicated reagents (n = 4 in each group). Mice were euthanized on day 14. Comparison of tumors with phosphate-buffered saline (control), carfilzomib (6 mg/kg), DC101 (40 mg/kg), or combined treatment with carfilzomib and DC101 for 2 weeks. (C) Subcutaneous tumor volume curves of Huh7 cells treated with the indicated reagents (n = 4 in each group, means ± SEM). (D) Subcutaneous tumor volume (upper panel) and weight (lower panel) in NOD/SCID mice treated with the indicated reagents on day 14 (one-way ANOVA test). (E) Representative immunohistochemistry staining images of FOXM1, AFP, and Ki-67 in subcutaneous tumors from NOD/SCID mice treated with the indicated reagents (scale bar = 100 μm). (F) Percentages of AFP-positive (left panel), FOXM1-positive (middle panel), and Ki-67-positive cells (right panel) counted three times from four independent immunohistochemistry staining images of subcutaneous tumors from NOD/SCID mice treated with the indicated reagents (means ± SEM, one-way ANOVA test).

  1. Please provide statistical analysis (p-values) for Figures 3A-C and 4.

We revised the manuscript accordingly.

Page 6 line 160.

Because FOXM1 activation in HCC was associated with serum AFP elevation, mitotic cell cycle activation, and poor prognosis, we tested whether FOXM1 could be a molecular target in AFP-positive HCC. We knocked down FOXM1 gene expression using siRNAs in Huh7 cells (Figure 3A, P < 0.05). Western blot showed the reduction of FOXM1 protein using the same condition (Figure 3B). Interestingly, FOXM1 knockdown resulted in a reduction of AFP gene expression in Huh7 cells (Figure 3C, P < 0.05). FOXM1 knockdown also inhibited cell proliferation (Figure 3D, P < 0.05), consistent with the reported role of FOXM1 as a regulator of the cell cycle.

Page 8 line 218.

Carfilzomib treatment at 20 nM for 24 h suppressed the viability of AFP-positive Huh7 and MT cells, but not AFP-negative Kami41 and KM cells (Figure 4D, P < 0.05).

  1. Lines 216-218: Provide reference for…. recent studies clearly showed the efficacy of ramucirumab, an anti-human VEGFR2 monoclonal antibody, for the treatment of AFP-positive advanced HCC in humans.

We added the references.

Page 9 line 237.

We planned to evaluate the effect of DC101 (anti-mouse VEGFR2 antibody) on AFP-positive Huh7 cell growth given that recent studies clearly showed the efficacy of ramucirumab, an anti-human VEGFR2 monoclonal antibody, for the treatment of AFP-positive advanced HCC in humans [8, 9].

Round 2

Reviewer 2 Report

Thanks for responding to most of my comments. These need to be addressed, however:

For major comment 1: Authors responded that FOXM1 mRNA expression data was mis-calculated and they re-analyzed the data. However, the data they have provided now is on linear Y-scale compared to log scale shown earlier. For AFP expression, data is shown on log scale. Does that also need to be re-calculated? Please clarify and provide more details on how the expression was calculated. In addition, the statistical comparisons need to be shown for different cell lines both for FOXM1 and AFP expression data.

For major comment 2: As requested, authors have included the tumor volume curves, however, the growth is shown only up to day 14. Authors responded that untreated mice start to die by day 42 so tumor volumes should be shown up to day 40 or so and data compared at that time point also (not only at day 14).

I do not agree with the statement on lines 254-256 that the combination treatment further reduces the number of AFP-positive, FOXM1-positive, or Ki-67-positive cells (compared to monotherapies). Authors may want to compare individual groups using Student’s t test.

Minor comments:

1.     Refer Fig. 5G on line 258.

1.     Re-number statistical analysis section in methods as 4.12.

Author Response

Reviewer 2

Thanks for responding to most of my comments. These need to be addressed, however:

We would like to thank the reviewer’s effort to improve our manuscript scientifically sound.

For major comment 1: Authors responded that FOXM1 mRNA expression data was mis-calculated and they re-analyzed the data. However, the data they have provided now is on linear Y-scale compared to log scale shown earlier. For AFP expression, data is shown on log scale. Does that also need to be re-calculated? Please clarify and provide more details on how the expression was calculated. In addition, the statistical comparisons need to be shown for different cell lines both for FOXM1 and AFP expression data.

We agree with the reviewer’s comment. We therefore re-calculated the AFP data. In both data, we utilized ΔΔCT method and now we set the Huh7 cells data as baseline. We also found that AFP gene expression was undetectable in Kami41 and KM cells. We revised Figure 4A and added the methods in materials and methods section.

Page 14 line 389.

Quantitation of genes expressed in cell lines relative to Huh7 cells was performed using the ΔΔCT method.

Page 9 line 226.

  • FOXM1 and AFP expression in Huh7, MT, Kami41, and KM cells (n = 3 in each group, means ± SEM, one-way ANOVA test).

For major comment 2: As requested, authors have included the tumor volume curves, however, the growth is shown only up to day 14. Authors responded that untreated mice start to die by day 42 so tumor volumes should be shown up to day 40 or so and data compared at that time point also (not only at day 14).

We apologize the presentation of confusing data. This is simply because we had experienced the difficulty in mimicking advanced stage HCC experiment in mice. Generally, for tumor xenotransplantation and drug effect evaluation experiment, we initiated the treatment when tumor volume reached 100~200mm3, and the treatment effect was much easier to evaluate. However, we tried to reproduce the advanced stage HCC condition in mice since anti-VEGFR2 antibodies are used in advanced stage AFP-positive HCC patients. We first tried to evaluate both tumor volumes and survival at the same time as initial experiment in vivo especially after the treatment (Figure 5A). We initiated the drug treatment at day 26 when average tumor volumes reached about 1,000mm3, 5~10 times larger than usual settings, and injected the last reagent at day 37 (day 11 after the initiation of the treatment). Unfortunately, when we tried to evaluate the tumor volumes at day 40, one mouse died (day 14 after the initiation of the treatment) and tumor was lost by cannibalism. The other mice also died at day 42 and later and we could not evaluate the tumor volume by the same reason. During the treatment period in this experiment, tumor volume result was almost similar to that presented in Figure 5C but the period was shorter and up to day 37 (day 11 after the initiation of the treatment). We therefore independently performed the tumor volume experiment using small numbers of mice and showed the results in Figure 5B and C. We appreciate it so much if the reviewer could accept our struggle in mice experiment mimicking the advanced stage HCC observed in human. We revised Figure 5A and C to include the time course information accordingly.

I do not agree with the statement on lines 254-256 that the combination treatment further reduces the number of AFP-positive, FOXM1-positive, or Ki-67-positive cells (compared to monotherapies). Authors may want to compare individual groups using Student’s t test.

We agree with the reviewer’s comment and revised the manuscript accordingly.

Page 10 line 255.

Furthermore, carfilzomib treatment alone, DC101 treatment alone, and their combination reduced the number of AFP-positive, FOXM1-positive, or Ki-67-positive cells (Figure 5E and F, one-way ANOVA test).

Minor comments:

  1. Refer Fig. 5G on line 258.

We revised the manuscript accordingly.

Page 10 line 257.

Western blot data confirmed the effect of carfilzomib, DC101, and their combination treatment on FOXM1 reduction (Figure 5G).

  1. Re-number statistical analysis section in methods as 4.12.

We revised the manuscript accordingly.

Page 15 line 466.

4.12 Statistical analysis